# Human-Centered Sensor Technologies for Soft Robotic Grippers: A Comprehensive Review

**DOI:** 10.3390/s25051508

**Published:** 2025-02-28

**Authors:** Md. Tasnim Rana, Md. Shariful Islam, Azizur Rahman

**Affiliations:** 1Bangladesh Council of Scientific and Industrial Research (BCSIR), Dhaka 1205, Bangladesh; 2Mechanical Engineering Department, Carnegie Mellon University, Pittsburgh, PA 15213, USA; sharifulmekuet@gmail.com; 3School of PCPM, Occupational Health and Safety/Ergonomics, City Campus, Royal Melbourne Institute of Technology, Melbourne, VIC 3000, Australia

**Keywords:** bio-robotics, gripper sensor, soft robotics, tactile sensing, sensor selection, PRISMA methodology, soft actuation

## Abstract

The importance of bio-robotics has been increasing day by day. Researchers are trying to mimic nature in a more creative way so that the system can easily adapt to the complex nature and its environment. Hence, bio-robotic grippers play a role in the physical connection between the environment and the bio-robotics system. While handling the physical world using a bio-robotic gripper, complexity occurs in the feedback system, where the sensor plays a vital role. Therefore, a human-centered gripper sensor can have a good impact on the bio-robotics field. But categorical classification and the selection process are not very systematic. This review paper follows the PRISMA methodology to summarize the previous works on bio-robotic gripper sensors and their selection process. This paper discusses challenges in soft robotic systems, the importance of sensing systems in facilitating critical control mechanisms, along with their selection considerations. Furthermore, a classification of soft actuation based on grippers has been introduced. Moreover, some unique characteristics of soft robotic sensors are explored, namely compliance, flexibility, multifunctionality, sensor nature, surface properties, and material requirements. In addition, a categorization of sensors for soft robotic grippers in terms of modalities has been established, ranging from the tactile and force sensor to the slippage sensor. Various tactile sensors, ranging from piezoelectric sensing to optical sensing, are explored as they are of the utmost importance in soft grippers to effectively address the increasing requirements for intelligence and automation. Finally, taking everything into consideration, a flow diagram has been suggested for selecting sensors specific to soft robotic applications.

## 1. Introduction

The development of robots made of flexible, malleable materials has led to a phenomenal boom in the rapidly developing subject of soft robotics, which dates back to the 1960s [1]. Inspired by the sensory and motor systems of biological species, these soft robots have several benefits over their rigid counterparts, including increased flexibility, compliance, and sensitivity.

There have been several noteworthy turning points in the development of sensor technology and soft robotics for the necessity of innovation. In 2010, Mannsfeld et al. [2] demonstrated the promise of flexible materials in sensing applications with the introduction of highly sensitive flexible pressure sensors utilizing micro-structured rubber dielectric layers, which was a good contribution to the field of soft robotics. Wang et al.’s seminal work from 2012 [3] on enabling energy conversion through the use of nanoscale triboelectric effects made steps toward smaller sensor technology. This study paved the way for the creation of self-powered tactile sensors for sustainable energy in portable electronics as this small sensor would consume less space in the system, and multiple sensor nodes could be used as they would take up less attaching space with minimum power consumption. A universal robotic gripper based on granular material jamming with Brown et al.’s groundbreaking work in 2010 [4] showed that employing soft materials in robotics applications is feasible. With the creation of a multi-gait soft robot, later advancements by Shepherd et al. (2011) showed the promise of soft robotics in completing difficult locomotion tasks [5] with the creation of a flexible piezoelectric nanogenerator built using a polyvinylidenefluoride-co-trifluoroethylene (PVDF-TrFE) thin film by Pi et al. [6]. The research conducted by Keplinger in 2013 was a significant milestone in the field of stretchable, transparent ionic conductors [7], which aided the foundation of the development of flexible and translucent ionic conductors. It played a crucial role in the advancement of soft robotics and prosthetic devices.

Most of the time, design concepts for soft robotics are relatively new. Here, Rus et al. (2015) made a substantial contribution to the field of soft robotics by providing a comprehensive overview of the design, construction, and control of soft robots and buildings [8]. Most important developments included in-depth analyses of soft robotic grippers, high-force soft printable pneumatics, and a thorough grasp of 3D printing methods for the fabrication of soft robots. Some relevant work by Jumet et al. [9], showing innovative compliant and underactuated robotic hands, multi-material three-dimensional-printed soft grippers, and an extensive data-driven review of soft robotics, respectively, represented further advancements in the field. In the year 2016, O’Brien successfully incorporated stretchy optical waveguides into a flexible prosthetic hand, thereby augmenting its optical functionalities [10]. The groundbreaking study conducted by Yi in 2019 on a customizable three-dimensional-printed origami soft robotic joint brought about a paradigm shift in the field of robotics by introducing innovative advancements in robotic locomotion [11]. In 2019, Lu conducted an investigation on the topic of “Pure PEDOT:PSS Hydrogels” with the aim of addressing concerns related to the purity of the material [12]. In the year 2020, Liu made substantial advancements in improving the durability of hydrogel adhesion, which is a crucial element in ensuring the longevity of soft robotics. Zhao’s contributions have had an indirect influence on the field of soft robotics, specifically in the areas of battery safety and energy density enhancement [13]. In 2021, Shen and Zhang presented novel underwater capabilities and mobility tactics inspired by biological systems [14]. The same year, Tan conducted research that emphasized the versatility of soft robotics in the context of aquatic situations [15]. Significant progress in soft robotics was demonstrated by recent developments in 2022 and 2023. These included improved controllability of soft robotic actuators through forward kinematics-based prediction methods, 3D-printed flexible electro-adhesion grippers, additively manufactured nano-mechanical energy harvesting systems, thin-film electronics on active substrates, and active control for sitting comfort. In the year 2023, Georgopoulou made significant improvements in the capabilities of soft robotics through the development of soft self-regulating heating elements [16]. Additionally, Gu demonstrated the significant contributions of this sector in the growth of neuroprosthetics [17]. Ionic liquid optoelectronics-based multimodal soft sensors were explored by Xu, who introduced a sophisticated sensor that combines the functionalities of ionic liquids with optoelectronics [18].

There are a number of articles that concentrate on sensors in general; nevertheless, there is limited writing dedicated to the development of sensors, especially for soft robotics. Some of the writings concentrate on specific application-based technology used in any specific soft robotic mechanism. Currently, a lot of work is being undertaken not only for medical purposes but also for military or commercially used products where soft robotic sensors play a vital role. Still, there are a lot of scopes for work in this field, but less documented work has been found on these soft robotic sensors. Moreover, a systematic approach to selecting types of sensors that should be used for any specific system is not clearly mentioned in the broader picture. This text seeks to provide a thorough compilation of advanced soft robotic sensors, offering readers an overview of their functions, intended applications, and the synergy between sensor technology and soft actuators. The goal of this writing is to compile and consolidate not only old but also the latest developments that have taken place in the field of soft robotics regarding sensor technology and suggest a systematic approach for selecting the sensor type for specific applications. This review paper is compiled in such a way that researchers and practitioners toward optimal sensor choices for soft robotics would have an overall view of soft robotic sensors. Also, its integration with modernized soft actuators for enhanced performance and diversified functionalities in soft robotic systems would be viewed in a broader aspect. This will be accomplished by providing details on appropriate sensor technologies and the respective purposes for which they are used.

## 2. Methodology

This systematic review was planned and implemented in 2024 and is mainly based on the PRISMA systematic screening process. Information and technology that were described in this research paper were searched for in the international databases of Web of Science, Google Scholar, and Scopus using the Litmap tool. The following keywords were used for our search: (A) in relation to sensors, we searched for bio sensors, tactile sensors, soft tactile sensor, and recent sensor developments; (B) related to robotics, we searched for soft robotics, soft robot sensor, robot sensor, sensors for soft robotic arm, and humanoid robot sensor; (C) for terms related to characteristics focused on output, we searched for compatibility of a sensor, multifunctionality, material requirement for fabrication, and surface condition of the sensor; and (D) for terms for the classification of sensors, we searched for sensor classification of robotic use, modern sensor classification, biosensor classification, and soft robotic sensor classification. General searching technique was used to combine keywords for database search. Following the conclusion of the search, a database of writers’ years and titles was constructed. Subsequently, duplicate and irrelevant items were meticulously screened and deleted. All irrelevant articles were removed from the search database.

### 2.1. Eligibility Criteria

#### 2.1.1. Inclusion Criteria

This writing used the following objectives or criteria to determine whether to include a study:(1)Only the peer-reviewed studies that have been published in the English language;(2)The study must involve soft robotic sensors used in various applications;(3)Studies that specifically show different kinds of sensors used in soft robotics;(4)Those scientific papers/articles that were published between 1960 and 2023 (July);(5)The literature search was restricted to journal papers, conference proceedings, books, reports, and relevant websites;(6)Newly developed sensors that are being used in commercial aspects but have yet to receive publications;(7)AI and data fusion in sensor technology that are physically the same sensor, but where data manipulation results in less computational power as it is also included as a sensor.

#### 2.1.2. Exclusion Criteria

Studies were excluded from the review for the following reasons:(1)There was no precise research population (for example, not specified or overly wide);(2)Not technically scientific articles, such as editorials or opinions;(3)Sensors that are not related to soft robotics;(4)Sensors that are too big and cannot be used as biosensors.

## 3. Soft Robotics and the Importance of Sensing Systems

The field of soft robotics is of great importance when considering the implementation of closed-loop control systems. Because soft robotics operates via the deformation of the robot body in response to differential stress, whether it originates internally or outside [19]. While undergoing deformation, it does not exhibit any predetermined shape change that would occur in the case of a rigid manipulator or robotic structure [20]. In this case, the primary challenge is accurately identifying the deformation or displacement inside a closed-loop system in a three-dimensional framework [21]. Soft robots depend on morphological computation, a phenomenon that is influenced by their physical configuration and material properties. This is in contrast to inflexible robots, which need precise control over their joints and limbs [22].

The challenges mainly depend on the identification of the state for soft robotics. The identification of soft robotics within a 3D framework poses challenges due to their non-rigid nature, which distinguishes them from conventional rigid structures [23]. So, the control of soft robotic flexible bodies is achieved through structural computation, primarily relying on the material’s inherent properties [24] and the matrix node that signifies its change. Soft robotic bodies face a significant challenge pertaining to the intricate modeling and design of dynamic control within a precise algorithm [25]. It is imperative to integrate sensors within the structure of soft robotic bodies in order to achieve precise sensing feedback, thereby facilitating the maintenance of critical control mechanisms [26]. The utilization of sensors facilitates the control and monitoring of the soft robot [27]. These sensors facilitate the monitoring of the shape and position of an object, enhancing its situational awareness through the implementation of a closed-loop control system [28]. While it is a necessity to ensure optimal performance, a soft robot necessitates a sensor system that exhibits a high degree of sensitivity and possesses excellent regulatory capabilities [29]. The process of collecting feedback from the sensor ultimately leads to the accumulation of a substantial dataset [30]. The evaluation and execution of this data set in field action would require a high-frequency processor [31]. The integration of sensors should be executed in a manner that minimizes the computational burden on the microcontroller during data processing. This would help researchers resolve numerous challenging issues encountered in practical contexts [32]. This is the reason why the sensor system in soft robotics is complex and critical to choose which type of sensor system the fabricator would use. As poor decisions will result in poor final performance, this could impact the data processing time during operation. The selection of an optimal sensor system for a particular task ultimately leads to the acquisition of a high-quality dataset while minimizing the burden on the microcontroller.

The sensing system plays a vital part in the operation of a closed-loop control system. The system should be built to effectively sense environmental conditions, as these conditions play a crucial role in determining the production material for soft robots [33]. Due to the prevalent use of soft and flexible materials in the fabrication of most soft robots, their responsiveness to environmental conditions can be very pronounced. There are additional parameters of significant importance in the selection of soft robotic sensors, as depicted in Figure 1. The incorporation of adaptation and the utilization of feedback from sensors are crucial in determining the likely shape and position of objects in the real world [34]. This feature serves as the fundamental component of a soft robotics sensing system, which must be taken into account alongside considerations of safety and the system’s ability to interact with humans [35]. Both factors are required because soft robotics are commonly used in environments that require gentle handling. The variable flexibility of soft robots allows for the achievement of several degrees of freedom (DoF), enhancing their controllability. This characteristic, together with their bio-inspired design, further enhances their applicability and overall functioning [36,37,38].

## 4. Degree-of-Freedom Actuation Systems in Soft Robotics

There are some jobs that grippers with one degree of freedom can only do, like changing the shape of an object in a single direction. A power source called the activation power decides whether the system will move or not. The system’s moving force can come from a pneumatic, hydraulic, magnetic, or other source [39]. Most of the time, a soft gripper with a single chamber opens or closes points to a single actuator that does a single task [40]. It usually has one main actuation point or device that lets it do a certain type of motion. In soft robotics, different types of motion systems are used. A model of the different actuation systems can be seen in Figure 2. This actuation system changes the path of the movement, which is one of the most important things to think about when picking a sensor [41].

## 5. Important Characteristics Required for Soft Robotic Sensors

Soft robotic sensors require unique characteristics that go beyond traditional metrics like accuracy, precision, sensitivity, resolution, repeatability, and minimal impact on measured quantities. Key qualities include compliance, flexibility, multifunctionality, sensor nature, surface properties, and material requirements as shown in Figure 3. Compliance ensures sensors adhere to the deformable nature of soft robots, allowing them to detect and respond to intricate deformations. Multifunctionality optimizes the design by detecting pressure, temperature, and strain simultaneously, streamlining the system and enhancing operational efficiency. Sensors must exhibit high sensitivity to pressure and shear forces, enabling real-time feedback and adaptive responses to environmental changes. Surface properties significantly influence sensor design, ensuring consistent and accurate sensing and interaction capabilities across diverse environments. Material requirements include flexibility, durability, and biocompatibility, which must endure deformations and strains without compromising sensing capabilities. Careful material selection is essential in sensitive settings like medical or biological applications to maintain compatibility and mitigate potential adverse effects.

### 5.1. Compliance/Flexibility

The preservation of compliance and flexibility inside the soft robotic body is crucial for the sensors utilized in soft robotics. In Figure 4 the compliance or flexibility relation with the sensor type is given. It is important for the modulus of the materials to be relatively low, often falling within the range of 10^5^ Pa to 10^9^ Pa, in order to align with the pliable composition with which they are incorporated [43]. Additionally, it is crucial that the integration of the sensor with the soft robot does not result in any discernible alterations in impedance across the entirety of the structure.

### 5.2. Nature of the Soft Robotic Sensor Surface

In order to facilitate smooth physical interaction with the soft robotic layers, it is imperative that the surface of the sensor possesses a soft texture. When the integration of the sensor layer occurs externally to the soft robotic component, it is imperative that the biomimetic functionalities of the robotic element remain uncompromised. So, while using sensors in soft robotics, the system could be used in sensitive medical-related work or other chemical laboratory work. For soft gipping, a three-finger silicon robotic gripper shown in Figure 5. In that case, the sensor surface plays an important role as it would also act as a surface of the system. The functionality of the gripper could be mishandled with the wrong sensor surface condition. So, the surface condition should ensure conformity, adhesion, and durability.

### 5.3. Material Requirements in the Construction of Soft Sensors

It is imperative that the sensor material possess stretchability, similar to the other constituent elements found in soft robotics. Polydimethylsiloxane (PDMS) elastomers and a variety of silicone elastomers are frequently employed in the construction of soft sensors [46]. In certain instances, the addition of impurities such as carbon black particles or a mix of composites and carbon nanotubes (CNTs) can be employed to augment conductivity in order to optimize sensing systems [46,47]. The selected material should also exhibit adequate fatigue resistance in order to endure anticipated cyclic stress [48]. The utilization of cost-effective materials is favored in order to minimize production expenses.

### 5.4. Multifunctional Sensor

Ideally, soft robotic sensors should possess the capability to perceive several parameters akin to the human skin’s capacity to discern distinct attributes, including force, temperature, shape, surface polish, and additional factors. The integration of many functions not only optimizes spatial utilization but also mitigates expenses related to employing distinct sensors for unique measurements. These kinds of multifunctional sensors are newly developed and introduced by specific applications in a functional format [49], where the user of this kind of sensor can have better feedback and response from a single point and can predict the correction or current condition of the system itself [50]. While designing such a multifunctional sensor, the designer can have better mobility in fabrication and easy access to position the sensor, as it would require a small space to consider the same physical sensor for the same feedback system [51].

## 6. Types of Sensors for Soft Robotic Grippers

Soft robotics is a very significant field of research as it has enabled many commercial aspects or benefits for practical applications. As they are used in more complex applications than traditional activities like simple movement or one degree of freedom work or simplicity in fabrication material [52]. Their specialty leads to diversified applications in the robotics field as increased dexterity and mechanical compliance of this kind of soft robotics come with the need of accurate control of position and shape. There are five types of sensing properties that a system would need, namely hearing, sight, smell, taste, and touch [53].

These sensing properties would also be required for any human being to understand the environment. For facilitating these sensing properties, there are a lot of methodologies for classifying the sensor types. Within the comprehensive classification of sensors for soft robotics, the tactile and force sensors have been very rapidly and widely used as robots’ capacities for tactile feedback and force detection. There are other factors, such as object property sensing, which could be the chemical composition or percentage of a material or other similar object property. To enhance clarity and facilitate a deeper understanding of sensor modalities and their methodology, a block diagram is provided in Figure 6. In Figure 7 sensor classification has been shown according to modalities.

### 6.1. Tactile and Force Sensing

One of the important factors of sensing is force sensing. The field of soft grippers has experienced notable progress regarding force sensing in both research and practical applications. One area of special interest is the development of tactile sensing technology for soft grippers, which allows for delicate interactions with the surrounding environment. This discovery is of significant importance as it effectively tackles the constraints associated with traditional hard grippers, which are not well-suited for operations in ambiguous and unorganized settings. Soft grippers that are endowed with touch sensory capabilities have enhanced cognitive capacities [56].

The soft grippers have the capacity to tactfully perceive, seize, and operate diverse items with a high degree of safety, owing to their sophisticated tactile sensing capabilities [57] shown in Figure 8. Moreover, the incorporation of force tactile sensors into soft grippers enables the manipulation of delicate objects without incurring any harm, owing to the use of efficient force feedback control techniques [58]. The utilization of tactile sensing technology in soft grippers is of utmost importance in order to effectively address the increasing requirements for intelligence and automation. During the initial phases of investigation into soft grippers, the primary emphasis was placed on the development of their structural design and actuation technologies in order to facilitate the execution of particular grasping tasks. For example, researchers have developed novel soft grippers that consist of several links and a system of pulleys operated by wire pairs [59]. These grippers have the ability to mold objects with different shapes in a gentle manner and maintain a consistent grasp [60]. An alternative methodology was employed, wherein a universal robotic gripper was developed utilizing granular material jamming. This gripper exhibited the ability to dynamically adjust its shape, hence ensuring consistent and reliable object manipulation. In a similar vein, a proposal was put up for a soft pneumatic robotic gripper that had a universal nature and the ability to adjust its effective length [61]. In Table 1 various tactile sensing techniques and their working principles are given along with their classification in Figure 9.

**Figure 8 sensors-25-01508-f008:**
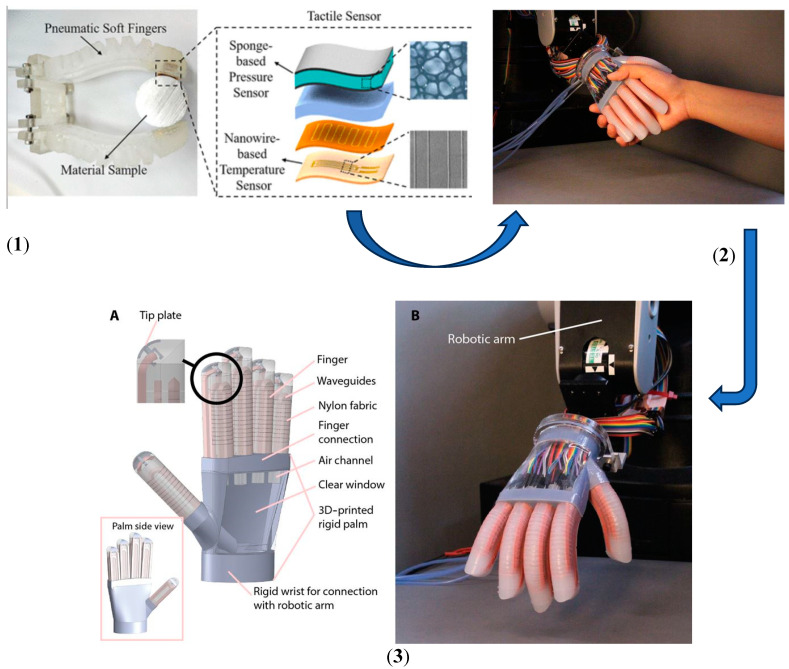
(**1**) Fabrication process and the raw material for force sensing. (**2**) Final fabrication and output. (**3**) Here (A). various parts of a soft robotic gripper, along with the sensor system; (B). full gripping system with sensor integration [62].

**Figure 9 sensors-25-01508-f009:**
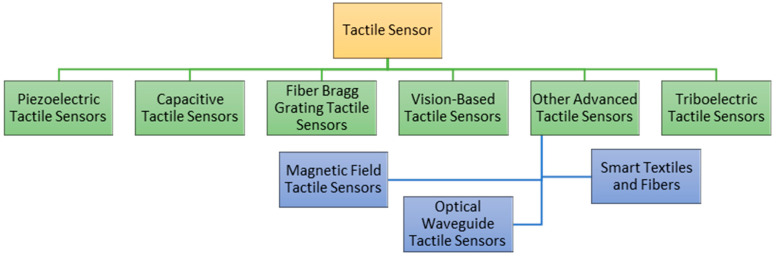
Classification of tactile sensor [63].

**Table 1 sensors-25-01508-t001:** Various tactile sensing techniques and their working principles, along with advantages and disadvantages.

Type of Sensor	Working Principle	Measurement	Advantage	Limitation	References
Resistive Ionic	Change in electrical resistance due to the movement of ions in a liquid or gel.	Liquid or gel level, concentration, and density.	Low cost and simple to use.	Not very accurate and can be affected by temperature.	[64]
Piezoelectric	Generation of an electrical charge when the sensor is subjected to mechanical stress.	Force, pressure, and acceleration.	High sensitivity and accuracy.	Expensive and fragile.	[65]
Piezoresistive	Change in electrical resistance due to mechanical stress.	Force, pressure, and acceleration.	Low cost and durable.	Not as sensitive as piezoelectric sensors.	[66]
Piezo-capacitive Strain	Change in capacitance due to mechanical stress.	Strain, pressure, and acceleration.	High sensitivity and accuracy.	Expensive and fragile.	[67]
Flexible Electronics	Use of flexible materials to create sensors.	A variety of measurements, including strain, pressure, temperature, and chemical concentration.	Lightweight and conformable to curved surfaces.	Not as durable as traditional sensors.	[68]
Capacitive Strain	Change in capacitance due to the change in distance between two electrodes.	Strain, pressure, and displacement.	High sensitivity and accuracy.	Can be affected by environmental factors such as moisture and dust.	[69]
Conductive Thermoplastic Resistive Strain	Change in electrical resistance due to the change in temperature of a conductive thermoplastic material.	Strain, temperature, and pressure.	Low cost and durable.	Not as sensitive as other strain sensors.	[70]
Optical Sensing	Use of light to measure various physical and chemical properties.	A variety of measurements, including strain, pressure, temperature, and chemical concentration.	Non-contact and can be used in hazardous environments.	Can be expensive and complex.	[71]
Strain-Sensitive Textiles and Fibers	Use of textiles and fibers to create strain sensors.	Strain, pressure, and movement.	Lightweight and conformable to curved surfaces.	Not as durable as traditional sensors.	[72]

### 6.2. Object Property Sensing

Soft robotics employs the utilization of object property sensing in order to engage with and manipulate their surrounding environment. A multitude of sensors are utilized to identify and quantify features of objects and the surrounding environment, hence facilitating intelligent exploration and responsive actions. Pressure sensing can be achieved through the utilization of force-sensitive resistors (FSRs) and capacitive pressure sensors. These technologies offer a comprehensive understanding of surface pressure by capturing subtle variations. Accurate temperature detection is of paramount importance for soft robots operating in a wide range of settings. For any given task to the robot, the gripper must act upon the object’s property; whether the object is soft or hard, it can differ in terms of being a liquid, solid, or semi-solid. The gripper surface can adapt to different objects by changing its material or surface condition based on the chemical properties of the object it interacts with. Medical robot grippers can also act as a crucial system where object property recognition would be challenging. Thus, the gripper must act according to the medical equipment or body parts that will be handled. For laboratory work in a nuclear radioactive zone, an operated soft robot must have a sensing system about the object’s property to clarify its output as per the safety guideline or allowable handle condition. To achieve this, the utilization of flexible thermocouples and thermistors is imperative since they offer highly exact measurements. In certain scenarios when direct touch is not feasible, infrared (IR) sensors have the capability to detect temperature without physical interaction. The utilization of chemical sensing enables soft robots to assess and react to their environment on a molecular scale. All the properties are summarized at Table 2.

Gas sensors are utilized for the purpose of identifying particular gases in order to monitor the environment and detect gas leaks. pH sensors, on the other hand, are employed to detect environmental chemicals, while biosensors are utilized for the detection of biological substances. The utilization of object property sensing in soft robotics enhances their ability to execute tasks with more precision and efficiency. This advancement allows them to successfully navigate intricate and ever-changing environments, whether it involves delicate object manipulation or intricate spatial navigation.

### 6.3. Proximity and Object Recognition Sensors

Proximity perception in human-centered robotics is used in two main categories: Applications of Type I (AT-I) and Applications of Type II (AT-II) [77,78]. AT-I involves the use of a delicate skin material to cover the links of a robotic manipulator, aiming to enhance safety measures and improve interaction functionality [79] shown in Figure 10. AT-II involves the integration of sensors into a robot gripper or hand to facilitate tasks related to grasping and exploring. The concept of sensitive skin with proximity awareness capabilities has been proposed to address the perception gap between humans and robots [80]. The ability of proximity perception to assess the operational environment of a robot is crucial for robotic solutions that comply with established norms and standards.

Proximity perception in human-centered robotics focuses on analyzing short distances between humans and robots, focusing on improving the quality of human–robot interaction and ensuring safety. Applications can be categorized into two main groups: those related to safety and human–robot interaction (HRI) or human–robot collaboration (HRC), and those associated with reshaping and grasping [82]. Automated behaviors facilitated by proximity sensors can be classified by their level of conceptual intricacy and their immediate impact on the robot’s movement [83]. An example of low-complexity behavior is the implementation of a safety stop mechanism, which is activated when a sensor signal surpasses a certain threshold, causing the robot’s brakes to engage [84]. In contrast, high-complexity behavior, such as object exploration, requires the ability to handle an object model and use a planner to carry out intentional exploration actions. Figure 11 shows the Integrated sensor system along with proximity sensor on robotic skin which can run the close loop sensing system on relative manner.

### 6.4. Slippage Sensor

Slippage sensing is a crucial aspect of soft robotics, addressing the challenges of the inherently compliant and deformable nature of soft robotic structures. It helps to prevent unintended relative motion between the robot and the objects it interacts with, ensuring a secure and controlled interaction [86]. Soft robotic systems use slippage sensing technologies to improve their grip and interaction with objects. These technologies help the robots adjust their movements dynamically to prevent slipping and maintain a stable grasp on objects [87].

Tactile sensors detect minute changes in contact forces, providing real-time feedback on the gripping force exerted on an object as shown in Figure 12. Computer vision and machine learning techniques are also used for slippage sensing in soft robotics, using cameras to monitor the relative positions of the robot and the object it is manipulating. Algorithms analyze these visual cues to identify instances of slippage, allowing the robot to adjust its grip in real time. Force and torque sensors integrated into joints or actuators also contribute to slippage sensing by providing continuous feedback on the forces applied during manipulation, allowing the robot to adjust its grip and movements accordingly.

### 6.5. Sensor Integration and Data Fusion

The integration of data and information is widely utilized within the field of soft robotics [89]. The objective is to amalgamate data from many sources in order to enhance its accuracy and efficacy. Soft robotics is a field that employs both physical and ethereal components to extract information from the surrounding environment or pre-existing data. Sensors are utilized to quantify the intensity of signals, while educated opinions leverage knowledge and skill to provide valuable insights, and databases make use of stored data [90]. Sources on soft robotics do internal evaluations of observations. This interpretation is employed by devices or decision-makers [91]. Soft robotics research focuses on information and data fusion, driven by commercial, geographical, and military concerns. Advancements in sensor technologies, fusion algorithms, hardware, and software enable real-time data fusion in applications such as automated target recognition, applied robotics, air traffic management, and weather forecasting.

Pragmatism emphasizes the growing importance of information integration, with defense data fusion being widely recognized [92]. Many soft robotic applications require data integration, including expert opinion aggregation, multi-target tracking, picture categorization, opponent location detection, autonomous robotics, transportation systems, and sensor technology. Information fusion is demonstrated in soft robotics programs like ADVANCE, AGVs, and PROMETHEUS. Well-structured multi-sensor systems improve soft robot performance, providing redundancy, timeliness, complementarity, and cost-effectiveness [93]. Soft robotics research aims to integrate various sensory modalities to improve object recognition and decision-making. Additional sensory mechanisms like humans can enhance soft robots’ abilities, enabling accurate item identification and safe communication in unclear settings [94]. Multimodal sensors are needed to collect physical and chemical data, with a focus on their sensing capabilities and architecture. Temperature sensors monitor and record environmental temperature, while olfactory sensors identify items by analyzing their chemical qualities. Monomodal sensors provide a lot of data, but object interpretation is difficult due to obstacles and poor lighting. Multimodal sensor data are needed to evaluate item properties and improve recognition.

In the field of soft robotics, there are two types of multimodal sensors: tactile sensors, which detect touch and pressure, and stretchable strain sensors, which measure deformation and stretch. The integration of sensing elements in a spatial manner facilitates the acquisition of many kinds of data using a single sensor. The integration of triboelectric and pressure sensors in a hybrid electronic skin enables the measurement of material properties. The three-layer stack multimodal tactile sensors are capable of measuring many parameters, including contact pressure, object temperature, ambient temperature, and object thermal conductivity. Convergent sensors integrate several sensing principles. Multimodal sensors are capable of quantifying many physical parameters, including pressure, tensile strain, and vibration. The utilization of hybrid systems incorporating triboelectric, piezoelectric, and piezoresistive mechanisms [95]. By integrating these techniques, the sensor is able to effectively perceive and measure surface roughness.

### 6.6. The Significance of Multimodal Sensors in Soft Robotics

Multimodal sensors are crucial in enhancing the functionalities of soft robotics by enabling them to acquire data from various sensory modalities, enhancing their ability to engage with the environment more efficiently. These sensors include touch, optical, temperature, and olfactory senses, allowing soft robotics to perform various functions such as item recognition, manipulation, and navigation in complex environments. Recognizing objects in soft robotics presents significant challenges due to the dynamic and unstructured nature of real-world situations. Monomodal sensors, while valuable, may limit the information they can convey about an item, making precise recognition difficult. Multimodal sensors offer a more holistic perspective by integrating data from multiple sense modalities, such as temperature, pressure, and thermal conductivity, enabling precise object recognition and decision-making. Multimodal sensors are used in various domains of soft robotics, including material perception, object recognition, environmental interaction, and human–robot interaction [96]. When equipped with these sensors, soft robots can perform tasks such as object manipulation, temperature variation detection, and navigation over complex terrains. Such kind of multimodal sensor has been shown in Figure 13. Total number of node would be 3 for the system with some protective fabrication layer of 5. These sensors are especially valuable in scenarios where soft robots are needed to replace humans in harsh or hazardous environments.

### 6.7. Fabrication of Soft Sensors in Microscale

It is well understood that the higher the number of sensors associated with a system of interest, the more data can be obtained, and more informed decisions can be made. Therefore, if sensors can be fabricated at a micro or nano level, thousands of devices can be obtained from a single wafer. Focusing on the tactile sensor category, microfabrication processes of several strain sensors are reported [98]. A group from the University of Maryland introduced a flexible MEMS capacitive strain sensor using a low-cost molding process with two designs (lateral and transverse combs) to test sensitivity based on strain direction [99]. Their microfabrication process starts with a 4″ Si wafer followed by deposition SiO_2_, which works as a hard mask in later steps. A mold has been prepared by etching Si; then, the mold is filled with cPDMS. Finally, devices are peeled off from the mold, and the mold is reusable. More specifically, the fabrication process flow they developed was as follows:

The fabrication process flow shown in Figure 14 is one of the several ways of fabricating a mems-based strain sensor. In the literature, we found various fabrication process flows to manufacture a sensor with many folds of miniaturization. The following table (Table 3) summarizes the recent attempts at sensor fabrication with typical feature sizes. This would help future sensor enthusiasts obtain a brief idea about the recent standing of sensor fabrication technology.

## 7. Sensor Selection Process Using All of the Parameters

Sensors are not a new thing for research. For appropriate control of any system, engineers use a sensor to obtain feedback in a closed-loop control system [100]. But while having a specific application and fabrication method for the system, there are some criteria for choosing which sensor should be used [101]. It is crucial to choose a suitable sensor system and technology. Makers could utilize single- or multi-sensor systems or can utilize different dimensions of a single sensing system. When choosing a sensor, it is important to first take into account all of the factors listed in Figure 1. Knowing the type of actuator to be used is also helpful. In Figure 2, we see numerous different actuation systems. One must think about the sensor material and fabrication processes while thinking about the actuation system, given that the sensor’s placement may be affected by fabrication processes. Then, decide on the best technology to employ for receiving feedback from the actuator. If it yields enough data, the final field test should produce satisfactory results. But if not, then a single sensor with many data segments would produce more accurate results. If this fails, additional sensors and data fusion will be introduced. This is the proposed model for selecting sensors for soft robotic applications so that researchers and engineers have a frame flow for selecting their sensors for specific projects, while material and fabrication complexity would play a vital role. The diagram is shown in Figure 15 below.

## 8. Conclusions

In this study, the realm of soft robotics, with a focus on sensors employed in soft robotic grippers, was explored. Flexible robots have gained superiority over traditional rigid structures because of their enhanced compliance and dexterity. Throughout our investigation, we underscored the critical role of sensing systems in soft robotics for achieving a precise control mechanism via feedback loops. A systematic approach is adopted to establish a convenient selection process.
○The parameters considered in the selection of soft robotic sensors have been reported, which are environmental condition, adaptability and feedback, safety and human interaction, dexterity and manipulation, control and autonomy, and bio-inspired functionality;○Soft robotic sensors require distinctive features that go beyond traditional metrics such as accuracy, precision, and sensitivity. The key characteristics include compliance, flexibility, multifunctionality, sensor nature, surface properties, and material requirements;○The categorization of sensor types for soft robotic grippers provided insights into tactile and force sensing, object property sensing, proximity sensing, and the integration of multimodal sensors. These sensor modalities facilitate soft grippers to interact intelligently with their environment, facilitating tasks ranging from delicate interactions to complex object recognition;○Acknowledging tactile sensing as one of the most important tactile sensors has been explored, including piezoelectric, piezoresistive, resistive ionic, piezocapacitive strain, capacitive strain, and optical sensing;○Multimodal sensors play a fundamental role in the field of soft robotics by facilitating the acquisition of diverse information pertaining to the surrounding environment and various objects;○The sensor selection process outlined in this study serves as a practical guide for engineers and researchers, emphasizing the importance of considering various factors such as application-specific requirements and fabrication methods. We aim to contribute to the advancement of this rapidly evolving field by proposing a comprehensive model for sensor selection in soft robotic applications.

## Figures and Tables

**Figure 1 sensors-25-01508-f001:**
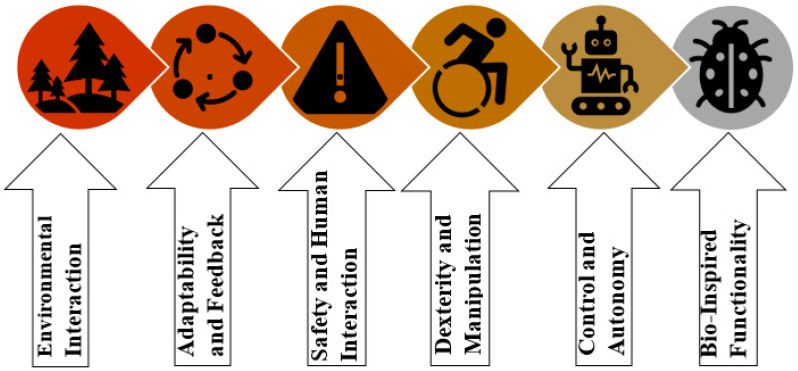
Consideration for sensors used in soft robotics.

**Figure 2 sensors-25-01508-f002:**
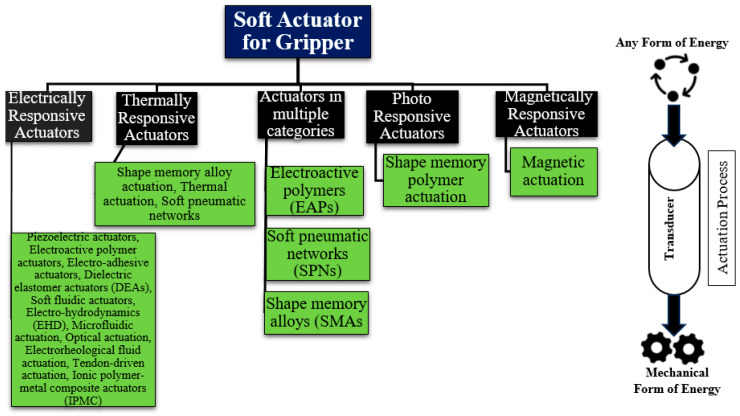
Classification of soft actuator in terms of gripper.

**Figure 3 sensors-25-01508-f003:**
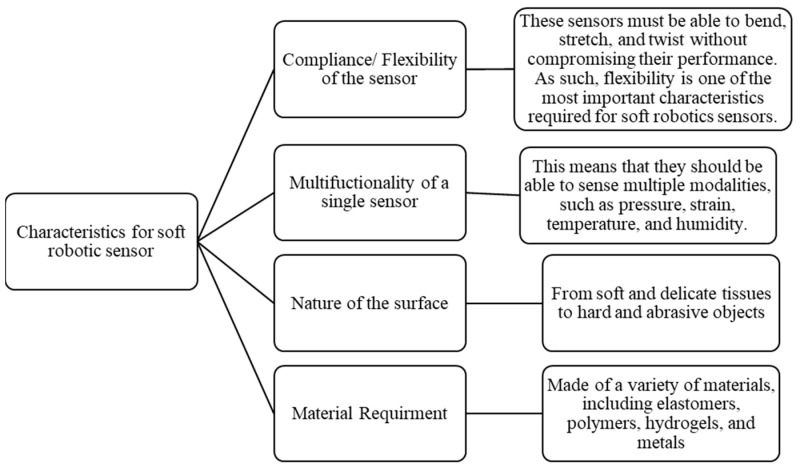
Characteristic trajectory for soft robotic sensor [42].

**Figure 4 sensors-25-01508-f004:**
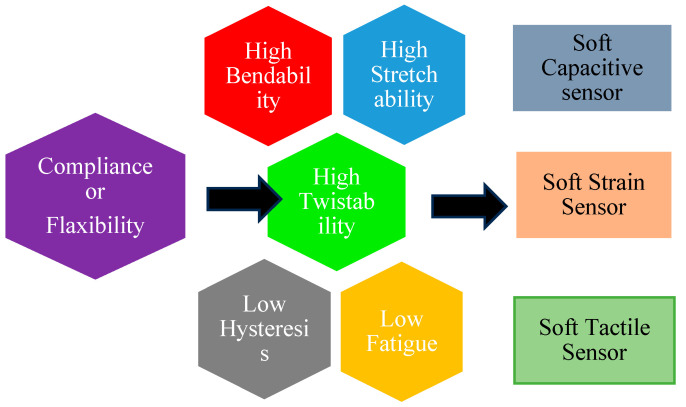
Factors affecting the compliance for soft robotic sensor [44].

**Figure 5 sensors-25-01508-f005:**
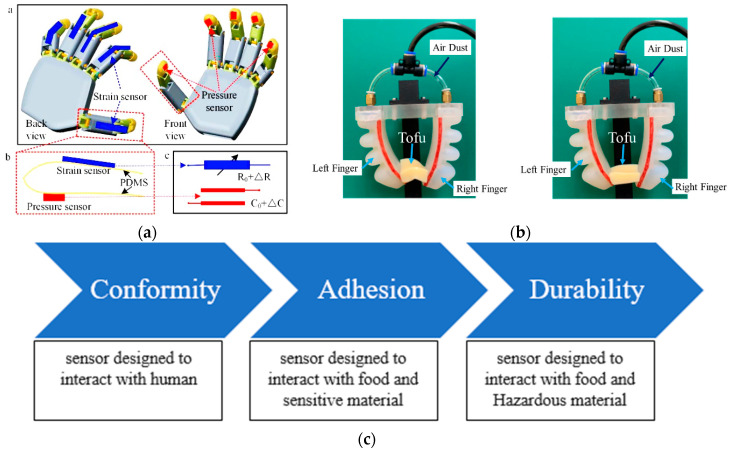
(**a**) Pressure sensor that could sense the pressure required to grab the surface, where a-b-c explains the methodology for sensing through resistive difference. (**b**) Basic fabrication of a gripper considering nature of surface of the material [45]. (**c**) Three basic parameters for conforming to the nature of the surface.

**Figure 6 sensors-25-01508-f006:**
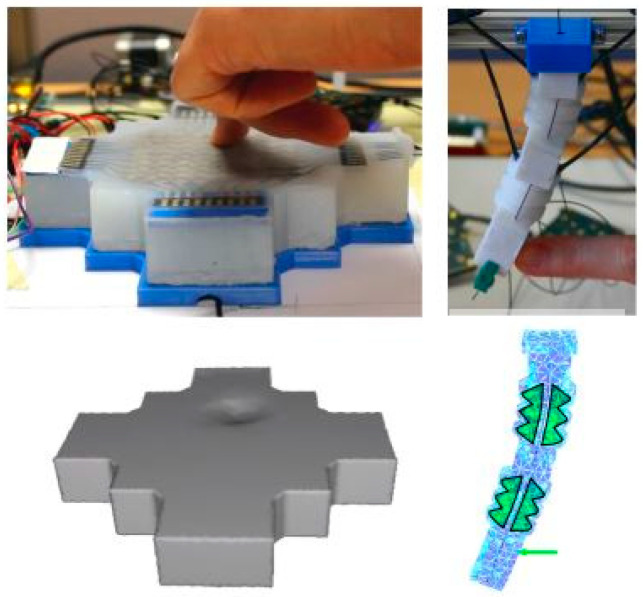
Two devices, namely a soft pad and a soft finger, have been introduced in this study. They incorporate both capacitive and pneumatic sensing. The utilization of capacitive sensing enables the identification of contact locations, while pneumatic sensing is employed for measuring deformation levels. The integration of sensor data is facilitated through a numerical model (FEM), enabling the estimation of applied forces and device deformation [54].

**Figure 7 sensors-25-01508-f007:**
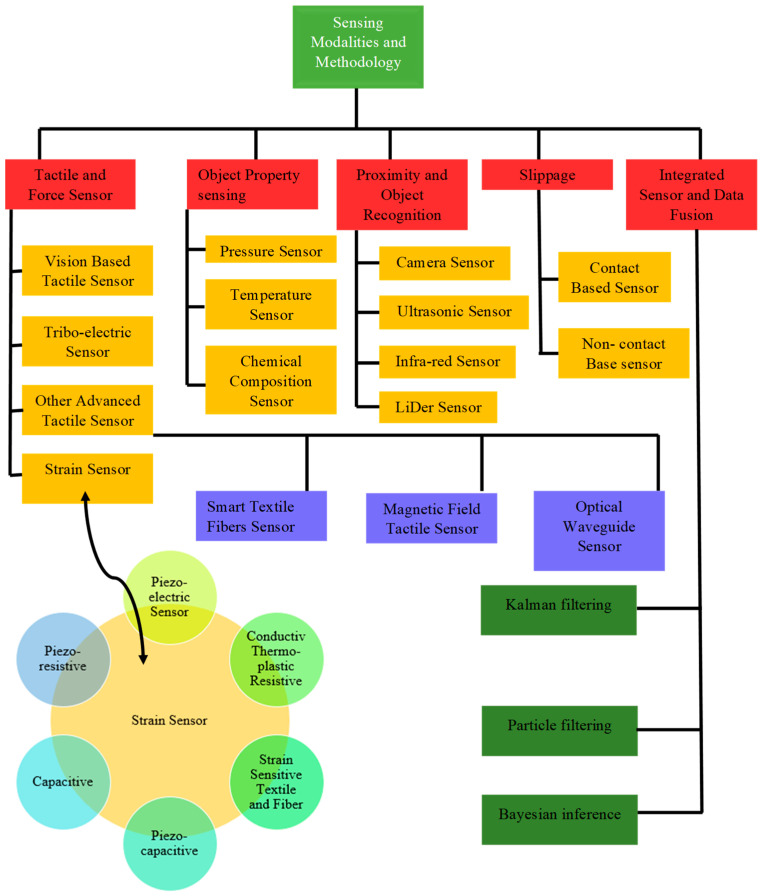
Classification of sense systems in terms of modalities [55].

**Figure 10 sensors-25-01508-f010:**
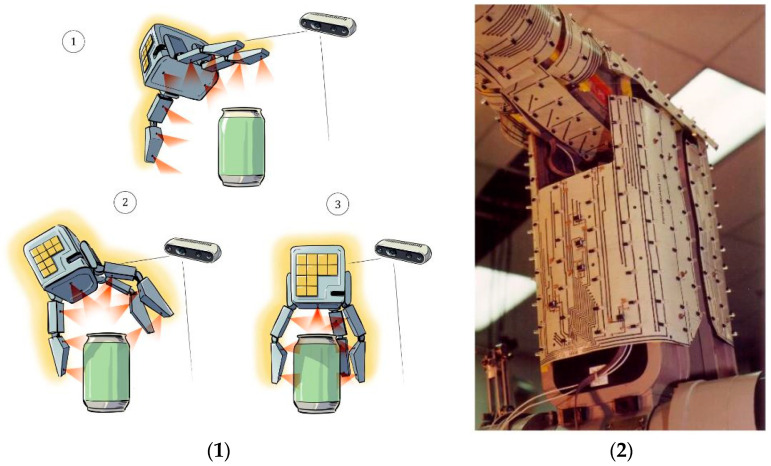
(**1**) Lidar sensor as a proximity approach in 3-stage object gripping from phase 1, 2 and 3. (**2**) Practical use of proximity sensors on robotic surfaces as a tactile properties without touching the body [81].

**Figure 11 sensors-25-01508-f011:**
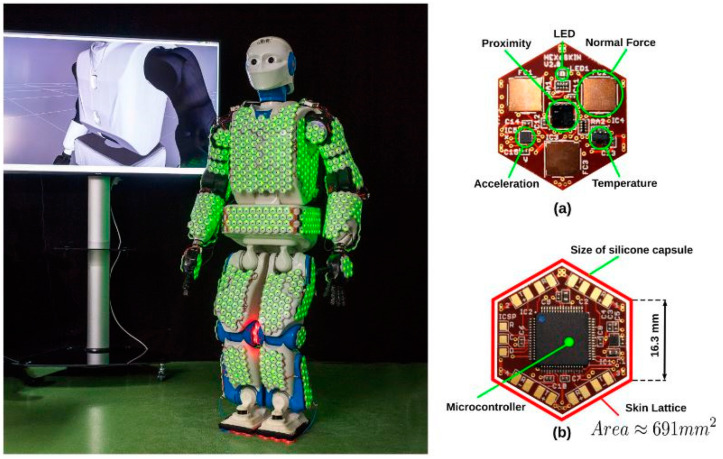
Integrated sensor system along with proximity sensor on robotic skin [85] where figure (**a**,**b**) shows the front and back side of the sensor.

**Figure 12 sensors-25-01508-f012:**
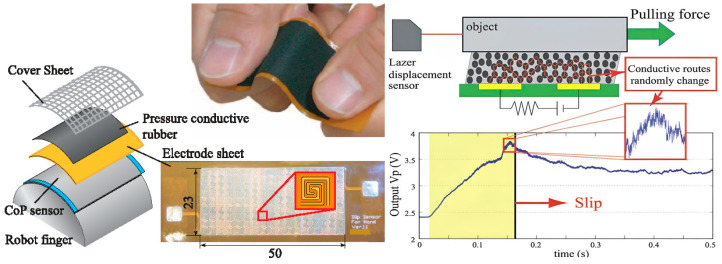
Advanced sensor for detecting initial slip in multi-fingered robot hands [88].

**Figure 13 sensors-25-01508-f013:**
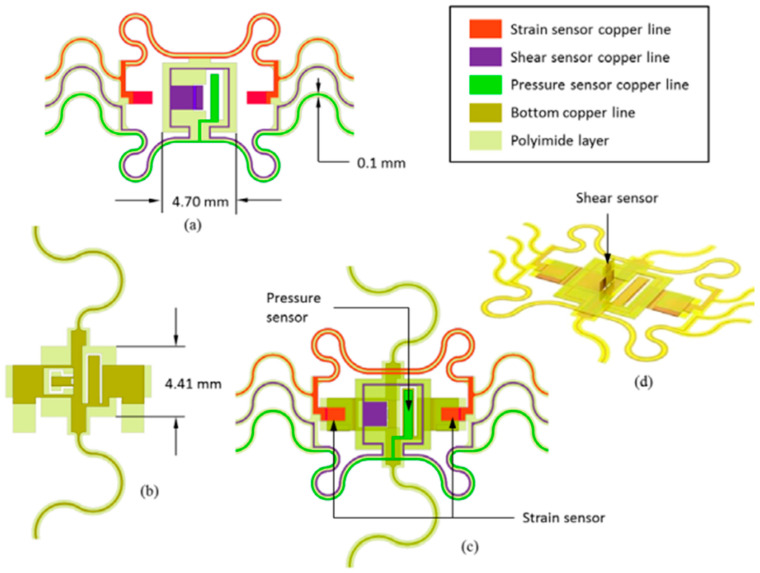
Sensor parts for a multimodal sensor system: (**a**) top layer, (**b**) bottom layer, (**c**) assembly, and (**d**) three-dimensional views [97].

**Figure 14 sensors-25-01508-f014:**
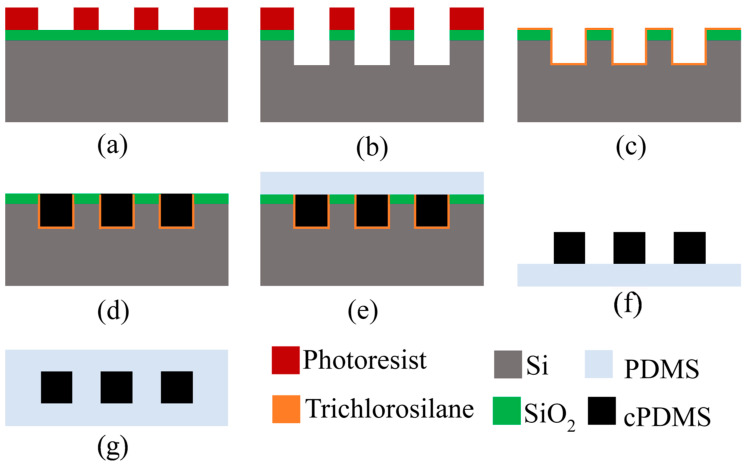
(**a**) Deposition of silicon dioxide and photoresist patterning (**b**) Perform a 6:1 buffered hydrofluoric acid (BHF) etch on the SiO_2_ layer followed by a deep reactive ion etching (DRIE) of the silicon wafer. (**c**) Deposition of trichlorosilane (**d**) Filling gaps with cPDMS, smooth out the surface, cure the material, and remove any excess cPDMS. (**e**) Application a PDMS coating. (**f**) Peeling off the devices from the mold. (**g**) Encapsulate the sensor areas with PDMS.

**Figure 15 sensors-25-01508-f015:**
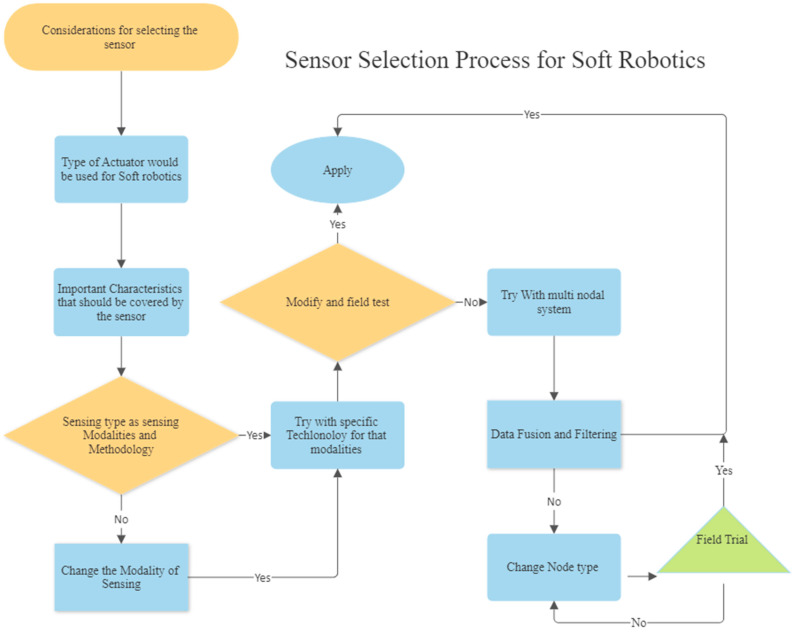
A non-conventional sensor selection process diagram.

**Table 2 sensors-25-01508-t002:** Types of properties a soft sensor could cover with their working principles.

Type of Property Sensing	Related Figure	Working Principle	References
Humidity	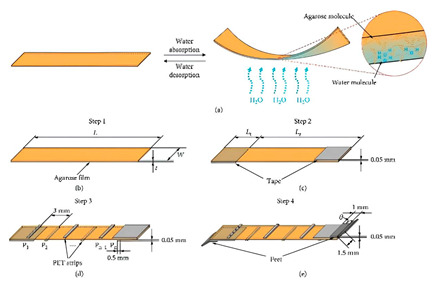	For properties related to humidity, the soft robotic sensor works by absorption and adsorption of water, which changes the permeability and impacts electrical current flow or conductivity by changed resistance.	[73]
Temperature	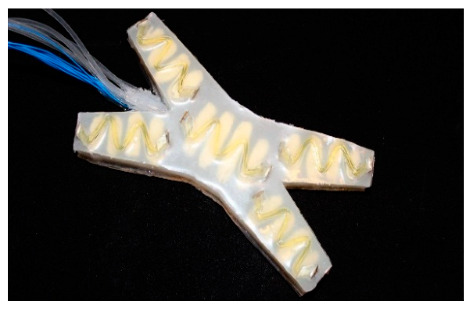	By changing the temperature conductivity, permeability, reference length increase, pneumatic pressure difference, etc., function could lead to electrical signals and produce results such as temperature.	[74]
Density	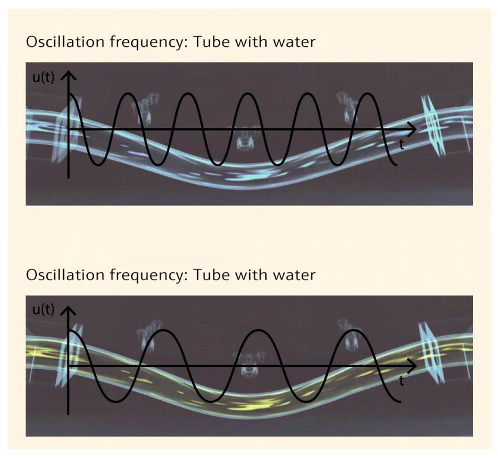	The resonator density method indirectly measures density by frequency. The liquid to be tested is placed in a resonance-vibrating tube. The oscillation frequency, which depends on liquid density and resonator rigidity, now indicates density.	[75]
Inflatability	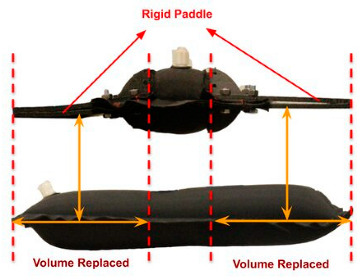	By using inflatable sensor, it provides integrated information collected from fiber optic distributed strain sensors woven into Vectran/Kevlar restrain layer, and it has foam layer shielding.	[76]

**Table 3 sensors-25-01508-t003:** Sensor type with fabrication process.

Sensor Type	Material Used	Feature Size (nm)	Fabrication Techniques (Dry Etching)	Applications
Resistive Sensors	SiO_2_, Polysilicon, Si_3_N_4_	10–100	RIE, DRIE	Pressure sensors, touch screens, microphones
Capacitive Sensors	SiO_2_, Si, Metals (Au, Al)	Nanometer Range	RIE, Plasma Etching	Proximity sensors, accelerometers, humidity sensors
Piezoelectric Sensors	ZnO, PZT	Tens to Hundreds	RIE, Chemical Etching	Vibration sensors, ultrasound imaging
NEMS Devices	Si, SiC, Si_3_N_4_	Nanoscale (device dependent)	DRIE, EBL + Plasma Etching	Microfluidic devices, gyroscopes
SPR Sensors	Metals (Au, Ag), Dielectrics (SiO_2_)	Nanoscale features (nanoholes, nanogratings)	EBL, FIB milling	Biosensing, chemical detection
Magnetic Sensors	Fe, Ni, Co alloys	1–100	Sputtering, Electroplating	Magnetic field sensors (compasses), medical imaging (MRI)
Gas Sensors	Metal oxides (e.g., WO_3_), Polymers	10–1000	Chemical Vapor Deposition (CVD)	Air quality monitoring, leak detection
Biosensors	Enzymes, Antibodies, Nucleic Acids	Varies (often larger than nano range)	Photolithography, Inkjet Printing	Medical diagnostics, environmental monitoring
Temperature Sensors	Platinum (Pt), Silicon (Si)	10–1000	Thin-film Deposition, Lithography	Temperature control systems, fire alarms

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
