# Peer review of "Human-Centered Sensor Technologies for Soft Robotic Grippers: A Comprehensive Review"

_sensors, 2025, doi:10.3390/s25051508_

Round 1

Reviewer 1 Report

Comments and Suggestions for Authors

This paper discusses challenges in soft robotics and their sensing systems. Some of the tactile sensors and their working principles are presented in this manuscript. However, based on my review, this manuscript requires significant revision. I strongly recommend a comprehensive revision addressing these issues as follows:

1.  The structure of the paper needs revision. A stronger focus on sensor technology is recommended.

2.  There are significant issues with the images used in the manuscript, including low contrast images, font formatting and arrangement, and confusing labels, which significantly reduce readability.

3.  Content shows confusion in sensor classification and working principles, which is particularly evident in Fig. 7 and Table 1. Moreover, Table 2 appears incomplete and uses ambiguous images, while Table 3 lists sensor fabrication techniques that are too limited.

4.  The conclusion lacks depth and merely reiterates the arguments presented in the article. Concerns have been raised that it may have been AI-generated. A thorough revision is necessary to provide a more insightful and original conclusion.

Author Response

Comments and Suggestions for Authors

This paper discusses challenges in soft robotics and their sensing systems. Some of the tactile sensors and their working principles are presented in this manuscript. However, based on my review, this manuscript requires significant revision. I strongly recommend a comprehensive revision addressing these issues as follows:

  1. The structure of the paper needs revision. A stronger focus on sensor technology is recommended.

Response 1: Thank you for your valuable feedback. In Chapter 6, the classification is primarily based on the modalities and methodologies used in sensing objects when designing the gripper platform. To clarify, by "modalities," we refer to the manner in which sensing occurs, focusing on how different aspects of the object are perceived for gripping. While there may be overlaps in the working principles or key functions, the classification in Chapter 6 aims to emphasize the specific modalities relevant to the object being gripped. This approach provides a clearer path for selecting the appropriate sensor in practical applications.

That’s why, although tactile sensors are mentioned in relation to both slip detection and other aspects, the modalities for selecting these sensors differ in each case. Despite the similarity in working principles, the way these sensors are applied varies based on the particular object and gripping requirements.

I hope this clarifies our rationale behind the classification, and we welcome any further suggestions to improve the manuscript. Chapter 3 explains “Soft Robotics and the importance of sensing system” here the main focus of discussion is how a bio robotics gripper dependent on its sensor feedback and this kind of feedback system would eventually help the system with respect to gripper “Degree of Freedom Actuation Systems in Soft Robotics” which has been discussed in the next 4th chapter. So, how we would classify this sensor? – the answer is modalities which is discussed at chapter 5. So, the justification of a new approach of bio robotics gripper sensor classification has been introduced focusing on modalities. So, chapter 3 to chapter 5 are closely inter connected. Maybe we have failed to explain the linkage, and hopefully do that at the corrected manuscript.

  1. There are significant issues with the images used in the manuscript, including low contrast images, font formatting and arrangement, and confusing labels, which significantly reduce readability.

Response 2: Thank you for your comments. We’ve updated all of our diagram for proper readability. But for cited reference images, we’ve used the source file from zip latex format. It’s difficult for us to have clearer image unless the cited paper provides them accordingly. We’re attaching our updated diagram file with the attachment.

  1. Content shows confusion in sensor classification and working principles, which is particularly evident in Fig. 7 and Table 1. Moreover, Table 2 appears incomplete and uses ambiguous images, while Table 3 lists sensor fabrication techniques that are too limited.

Response 3: Thank you for your valuable feedback. In Chapter 6, the classification is primarily based on the modalities and methodologies used in sensing objects when designing the gripper platform. To clarify, by "modalities," we refer to the manner in which sensing occurs, focusing on how different aspects of the object are perceived for gripping. While there may be overlaps in the working principles or key functions, the classification in Chapter 6 aims to emphasize the specific modalities relevant to the object being gripped. This approach provides a clearer path for selecting the appropriate sensor in practical applications.

It is true that Table 2 is incomplete; however, it aims to focus on "property" based sensor systems and their working principles. When selecting figures, we prioritize using reference images rather than creating conceptual illustrations. The availability and relevancy of sensor images are largely determined by the source files, and we rely on them to ensure accuracy of information from peer review journals.

  1. The conclusion lacks depth and merely reiterates the arguments presented in the article. Concerns have been raised that it may have been AI-generated. A thorough revision is necessary to provide a more insightful and original conclusion.

Response 4: Thank you very much for your concern. We always care for the originality of work. We’ve read through the conclusion and the writing is totally by human and Ai check has been done and the finding is that, 100% of the work from conclusion are human written.  And we request to make the AI check for further clarification.

Reviewer 2 Report

Comments and Suggestions for Authors

1. The authors of this paper have reviewed the sensor technology for soft robot grippers, but the content of sensor types is not better classified in my opinion. In  Chapter 6, sensor classification is a bit messy. As we know, the tactile force sensors summarized in Chapter 6.1 already contain the later content of object feature recognition, slip sensors. Please reconsider the sensor classification.

2. Please check the content from Line 131 to Line 149

3. What is the logical relationship between the contents of Chapters 3, 4 and 5 of this paper?

4. Figure 4 and Figure 14 are not clear.

5. In 5.2, the authors just suggest the properties that a surface sensor for a soft robotic gripper should have. It would be better to give examples of what methods or sensor structures should be used to achieve this goal.

6. Tactile pressure and temperature are mentioned several times in the multimodal sensors presented in 6.6, but the picture lists only pressure and strain sensors in Fig13. Could the authors give more examples of multimodal sensors that can detect force, temperature and humidity at the same time, so that they can better correspond to what is presented in text of 6.6.

7. Figure 8. (4) and Figure 14. (a) and (b) lack explanations.

8. References [13], [16], [41], etc. are missing journal name, and [34] is missing doi, please check the references carefully.

Comments on the Quality of English Language

Minor editing of English language required

Author Response

Comments and Suggestions for Authors

  1. The authors of this paper have reviewed the sensor technology for soft robot grippers, but the content of sensor types is not better classified in my opinion. In Chapter 6, sensor classification is a bit messy. As we know, the tactile force sensors summarized in Chapter 6.1 already contain the later content of object feature recognition, slip sensors. Please reconsider the sensor classification.

Response 1: Thank you for your valuable feedback. In Chapter 6, the classification is primarily based on the modalities and methodologies used in sensing objects when designing the gripper platform. To clarify, by "modalities," we refer to the manner in which sensing occurs, focusing on how different aspects of the object are perceived for gripping. While there may be overlaps in the working principles or key functions, the classification in Chapter 6 aims to emphasize the specific modalities relevant to the object being gripped. This approach provides a clearer path for selecting the appropriate sensor in practical applications.

That’s why, although tactile sensors are mentioned in relation to both slip detection and other aspects, the modalities for selecting these sensors differ in each case. Despite the similarity in working principles, the way these sensors are applied varies based on the particular object and gripping requirements.

I hope this clarifies our rationale behind the classification, and we welcome any further suggestions to improve the manuscript.

  1. Please check the content from Line 131 to Line 149

Response 2: Thank you for your comment. From Lines 131 to 149, we discuss the inclusion and exclusion criteria followed during our literature review and data collection, adhering to the PRISMA systematic screening process. I acknowledge that the alignment was incorrect in the initial submission, and I will ensure it is properly corrected in the next submission.

  1. What is the logical relationship between the contents of Chapters 3, 4 and 5 of this paper?

Response 3: Chapter 3 explains “Soft Robotics and the importance of sensing system” here the main focus of discussion is how a bio robotics gripper dependent on its sensor feedback and this kind of feedback system would eventually help the system with respect to gripper “Degree of Freedom Actuation Systems in Soft Robotics” which has been discussed in the next 4th chapter. So, how we would classify this sensor? – the answer is modalities which is discussed at chapter 5. So, the justification of a new approach of bio robotics gripper sensor classification has been introduced focusing on modalities. So, chapter 3 to chapter 5 are closely inter connected. Maybe we have failed to explain the linkage, and hopefully do that at the corrected manuscript.

  1. Figure 4 and Figure 14 are not clear.

Response 4: Thank you very much for the notification. Attachment  has the updated clear diagram. I would add them with the updated manuscript also.

  1. In 5.2, the authors just suggest the properties that a surface sensor for a soft robotic gripper should have. It would be better to give examples of what methods or sensor structures should be used to achieve this goal.

Response 5: Thank you for your concern. As in chapter 6 a elaborate explanation of sensor working method has been explained, we tried to keep chapter 5 small. As per the recommendation we would update that at the current version.

  1. Tactile pressure and temperature are mentioned several times in the multimodal sensors presented in 6.6, but the picture lists only pressure and strain sensors in Fig13. Could the authors give more examples of multimodal sensors that can detect force, temperature and humidity at the same time, so that they can better correspond to what is presented in text of 6.6.

Response 6:  Thanks for your recommendation. Change has been made mentioning multimodal sensors.

  1. Figure 8. (4) and Figure 14. (a) and (b) lack explanations.

Response 8:  Thank you for the notification. The explanation has been updated.

  1. References [13], [16], [41], etc. are missing journal name, and [34] is missing doi, please check the references carefully.

Response 8: Updated

Round 2

Reviewer 2 Report

Comments and Suggestions for Authors

I think this manuscript was sufficiently revised along the comments.

Comments on the Quality of English Language

Minor editing of English language required. Please check the manuscript carefully.